# Impact of COVID-19 on Short- and Medium-Term Prescription of Enteral Nutrition in the General Population vs. Older People in the Community of Madrid, Spain

**DOI:** 10.3390/nu14193892

**Published:** 2022-09-20

**Authors:** Carolina Luque Calvo, Ángel Luis Mataix Sanjuan, Ángel Candela Toha, Nilda Martínez Castro, María Rosario Pintor Recuenco, José Luis Calleja López, José Ignacio Botella-Carretero, Francisco Arrieta Blanco

**Affiliations:** 1Farmacia, Investigación Ramón y Cajal, Hospital Universitario Ramón & Cajal, 28034 Madrid, Spain; 2Responsable Sistemas Información Farmacia, Subdirección General de Farmacia y Productos Sanitarios Comunidad Autónoma de Madrid, Paseo de la Castellana, 280, 28046 Madrid, Spain; 3Department of Anesthesia & Reanimation, Hospital Universitario Ramón & Cajal, 28034 Madrid, Spain; 4Department of Hospital Pharmacology, Hospital Universitario Ramón & Cajal, 28034 Madrid, Spain; 5Department of Internal Medicine, Hospital Universitario Ramón & Cajal, 28034 Madrid, Spain; 6Department of Endocrinology & Nutrition, Hospital Universitario Ramón & Cajal, 28034 Madrid, Spain

**Keywords:** COVID-19, nutritional therapy, enteral nutrition, costs, older people, Community of Madrid, Spain

## Abstract

We aimed to analyse the impact of COVID-19 during 2020 and 2021 on the prescription of enteral nutritional support and its expenditure in the Community of Madrid, Spain, compared to pre-pandemic data from 2016 in the general population vs. elderly. We analysed official electronic prescriptions of all public hospitals of the Community of Madrid. The population over 75 years of age have the higher prescription of nutritional supplements (*p* < 0.001 vs. other age groups), with no differences between the 45–64 age group compared to the 65–74 age group (χ^2^ = 3.259, *p* = 0.196). The first wave of COVID-19 or the first time there was a real awareness of the virus in Spain is similar in a way to the first peak of prescription of enteral nutrition in March 2020. The second peak of prescription was observed in the over 75 age group in July 2020, being more pronounced in December 2020 and March–April of the following year (F = 7.863, *p* = 0.041). The last peaks correspond to summer 2021 and autumn of the same year (*p* = 0.031—year 2021 vs. 2020, *p* = 0.011—year 2021 vs. 2019), where a relationship between increased prescription of enteral nutrition and COVID-19 cases is observed. High-protein and high-calorie dietary therapies were the most prescribed in patients with or without diabetes. All of this entailed higher cost for the Community of Madrid. In conclusion, COVID-19 significantly affected the prescription of nutritional support, especially in the population over 75 years of age.

## 1. Introduction

COVID-19 is an infectious disease whose responsible pathogen is the SARS-CoV-2 virus, whose name was coined by the World Health Organization (WHO) to avoid confusion with the SARS virus of 2003. Both this new virus and the disease it causes were unknown before the outbreak in Wuhan (China) in December 2019. SARS-CoV-2 is an RNA virus belonging to the beta-coronaviridae [1] family. It has been found to cause severe pneumonia and acute respiratory distress syndrome (ARDS) with a significantly high mortality rate. Due to the rapid spread of this infection with global consequences, on 11 March 2020, the WHO declared COVID-19 a pandemic and called on countries to take appropriate action to stop its spread [2,3].

In Spain, since the first case of COVID-19 was confirmed in the country, 6,128,902 cases, 447,287 hospitalizations and 89,319 deaths from COVID-19 have been reported to RENAVE. From 10 March 2020 to 28 December 2021, estimations obtained with MoMo, the monitorization system of daily mortality englobing all kinds of causes, indicate that there have been 100,997 additional deaths from all causes nationwide [4].

Data on the effect of COVID-19 on the prescription of nutritional support, including enteral nutrition, are very scarce [5]. It is known that age, presence of comorbidities, immunosuppression or malnutrition are poor prognostic factors in patients with COVID-19 [6,7]. Considering that nutritional status is a relevant factor influencing the evolution of patients with COVID-19 [8,9,10,11], its approach in cases of malnutrition plays a key role both in the evolution of the disease and in the reduction of complications in patients who have suffered from COVID-19.

It is increasingly apparent that nutritional care, including identification of nutritional risk and use of nutrition support, should be a fundamental part of management for these patients. COVID-19 illness negatively impacts nutritional status on many levels; increas-ing nutritional requirements induced by pyrexia, sepsis, dyspnoea, and reducing nutri-tional intake due to excessive coughing, dysphagia, dysgeusia, chronic fatigue, poor appe-tite and food access issues. The detection and management of malnutrition in patients with COVID-19 is therefore of fundamental importance [12].

Hyperglycemia is commonly associated with adverse outcomes especially in patients requiring intensive care unit stay. Data from the COVID-19 pandemic indicates that individuals with diabetes appear to be at similar risk for COVID-19 infection to those without diabetes but are more likely to experience increased morbidity and mortality [13].

A previous study in which prescription was also analysed and compared to the impact of COVID-19 showed that there was a link between the virus and the increase in prescription of these products [14]. Further research with a broader view from previous years was needed to check the tendency that the prescription followed, and this was compared to the impact of COVID-19.

The aim of this study is to analyse the impact of COVID-19 from the beginning of the pandemic until the end of 2021, comparing it to the variation in prescription against pre-pandemic years since 2016. The target population comprises elderly patients who have been the most affected during this period and shows the difference compared to the general population [15]. Additionally, the different formulas for enteral nutrition will be discussed (diabetic, hyperproteic or hypercaloric) as these patients have a special need for supplementation due to their malnutrition that stems from the virus [16] and the high risk of diabetes in COVID-19 patients [17].

## 2. Materials and Methods

A descriptive and comparative analysis of the prescription of enteral nutrition products in public hospitals in the Community of Madrid during 2020 and 2021 has been carried out. The data have been analysed against the prescription trend since 2016. The data were derived from prescriptions of the hospitals of the National Health System. The data source used was the System for Analysis of the pharmaceutical provision of the Community of Madrid (Farm@drid), a population-based database of official medical prescriptions dispensed in the pharmacies of the Community of Madrid and billed from the health expenditure of the Community of Madrid. The following variables were obtained from this database: number of containers and amount by national code and age group, dispensed to patients prescribed enteral nutrition support.

The amount of enteral nutrition prescribed is measured by the containers. These are the bottles in millilitres or milligrams containing the enteral formula with the different substances: protein, carbohydrates, fibre, special containers for diabetes and some other enteral products that are not within the scope of this study. For the different formulas that could be used, the study was focused on supplements that were recommended by different guidelines over the world, such as the Society of Parenteral and Enteral Nutrition (ASPEN) and the European Society for Clinical Nutrition and Metabolism (ESPEN). These guidelines suggest a different approach depending on the patient and their possible needs (if they have other morbidities, for example). The approach consisted of the adjustment of their protein, lipid and/or energetic need [18].

In the age distribution of the data, the population was distributed into the following age groups: 15–44, 45–64, 65–74 and over 75 years.

Data from the Spanish Institute of Statistics (INE) have been verified to compare the increase in COVID-19 cases and admissions to hospital with the increases in enteral nutrition prescription; therefore, Figure 1 shows the increase in COVID-19 cases in Madrid during 2020–2021 [4].

Data extraction from computerized records was anonymized and did not include variables that could identify patients, so the study respects their confidentiality. Data are expressed as total absolute numbers. For statistical analysis, mean and standard deviation or proportions per category were used as detailed in the results. For comparison of means between different groups, a one-way ANOVA test with Tukey’s post hoc test was used. In the case of before-after comparisons between years, a repeated measures ANOVA was used. For the comparison of proportions, the chi-square test with Fisher’s correction where necessary was used. The analysis was performed with the SPSS 18.0 statistical package (SPSS Inc., Chicago, IL, USA), for a significance level of *p* < 0.05.

## 3. Results

The prescription of diet therapeutic containers over the years 2020 and 2021 fluctuated as COVID-19 infection changed [4].

Regarding the number of containers prescribed by age group (Figure 1), it can be seen that the 15–44 age group was prescribed the least (*p* < 0.001 with respect to the rest of the age groups) and those over 75 years of age were prescribed the most nutritional supplements during this period (*p* < 0.001 with respect to the rest of the age groups), with no differences found between the 45–64 age groups compared to the 65–74 age groups (χ^2^ = 3.259, *p* = 0.196). The first increase in prescription is seen in March 2020, as in the first wave of COVID-19, for every group older than 45 years. The second increase observed in the over-75 age group corresponds to July 2020, being more noticeable in December 2020 and March–April of the following year (F = 7.863, *p* = 0.041). The last peak belongs to summer 2021 and autumn of the same year (*p* = 0.031—year 2021 vs. 2020, *p* = 0.011—year 2021 vs. 2019). The containers prescribed during 2019 are also included in Figure 1 to see the evolution of the prescription in the year before the pandemic hit every country.

If prescription since 2016 is compared, a downward trend of this type of nutritional products could be seen, a trend that has been impacted and has suffered an increase coincidentally with the entry of COVID-19 in 2020 (*p* < 0.05—year 2021 vs. 2020, 2019 and 2018; *p* = 0.071—year 2021 vs. 2017, *p* = 0.256—year 2021 vs. 2016) (Figure 2).

In an analysis by type of diet therapeutic, pitting hyperproteic against normal protein, the trend since 2016 has been higher consumption of hyperproteic diets, which has continued even during the pandemic (*p* < 0.01 years 2021 and 2020 compared to previous years; *p* < 0.01—year 2021 vs. 2020) (Figure 3).

Table 1 also shows prescription by type of dietary therapy (hyper- or normal protein), as well as calorie and fibre intake in more detail. It is observed that the trend in prescription has been similar, with a change when the virus started to be noticed, where an increase in the prescription of complete oligomeric hyperproteic formulations (almost double) is observed, with a reduction in the prescription of normoproteic formulations. Likewise has happened with hyperproteic hypercaloric polymeric complete formulations with and without fibre, as well as with hypercaloric normoproteic complete formulations with and without fibre. Table 2 shows more clearly which diets had the greatest economic impact during those years and with the advent of COVID-19.

Finally, if we analyse diabetic diet therapeutic products, we can see the downward trend in normal protein (normoproteic diets) compared to the sharp rise in hyperproteic diets, which seemed to remain stable until the arrival of the virus, where an increase can be seen in 2020 and 2021 (*p* < 0.01 with respect to the previous years) (Figure 4).

In addition, the increase of this type of hyperproteic dietotherapeutic product can be seen in the over-75 age group compared to normoproteic diets and compared to the other age groups (Figure 5).

## 4. Discussion

In the present study, and as a continuation of the analysis carried out during 2020 in the first phase of the pandemic [14], we found that the maximum prescription of enteral support in the Community of Madrid was during 2021, which had previously peaked in 2020 due to the COVID-19 infection and with the declaration of the pandemic by the WHO [19]. Although the design does not allow us to assess the direct causes of this growth, we consider that it may be due to several factors, being the rise in the number of patients admitted to hospitals, especially at the beginning of the first wave of the pandemic, one of the most plausible causes, as patients were observed to have severe inflammation and illness. It has been reported that during the first quarter of 2020 in Madrid, the preliminary seroprevalence of SARS-CoV-2 was 11.3%, being the region with the highest mortality due to COVID-19 in Spain and among the regions most affected by the pandemic in the world at that time [2,19]. The role of nutrition and its support as a therapy to avoid the risk of malnutrition continues to be a common strategy to fight COVID-19 [20,21,22], so the increase in prescription would be justified during the pandemic years.

When analysing the prescription of nutritional support over the two years of the pandemic, a significant increase in the number of containers is observed during the year 2021 in all groups, coinciding with the contagion in the successive waves and deconfinement of the country [4].

From 22 June 2020 to the end of 2021 (2nd to 6th epidemic periods), 5,873,050 cases of COVID-19 have been reported to RENAVE in Spain; 51.7% are women, and the median age of the cases is 38 years. The 40–49 age group is the most represented, with 16.7% of the cases, followed by the 20–29 age group, with 16.1%. Some 5.7% of the cases were hospitalised. As can be seen in the results obtained when analysing the consumption of nutrients, there is an increase in the consumption of these groups, as well as an increase in the age group over 75 [4]. Not only was age consistently associated with higher risks of hospitalisation and mortality, but an increased risk of death has also been reported in outpatient diagnosis of COVID-19 without hospitalisation [21].

By age group, case fatality increases from the age of 60 onwards, both among total cases and among hospitalized cases. The severity indicators analyses (percentage of hospitalizations, percentage of ICU admissions among inpatients) increase with age from 30 years onwards in all periods, except for ICU admissions, which decrease from the age of 70 years onwards. Case and in-patient case fatality rates increase from the age of 60 and 50 years, respectively. The evolution of these indicators by epidemic period suggests that there is a decrease in hospitalization of cases from the age of 40 onwards, as well as a decrease in case and inpatient case fatality rates from the age of 60 onwards in the last epidemic period compared to the previous ones. In absolute numbers, the number of hospitalized cases, ICU admissions and deaths among those aged 40 years and older declined considerably in the last two periods compared to the previous periods [4].

The observed increase in prescribed enteral nutrition from 2020 to 2021 may be, among other causes, due to the increase in malnutrition and need for enteral nutritional support in all age groups as infection increases in the population [23,24]. Patients who are hospitalized with COVID-19 have a higher prevalence of malnutrition than their hospitalized counterparts without COVID-19 [25]. The parallelism of the prescription of containers and healthcare expenditure is notable, with no significant differences in total costs from 2016 to 2021, as it was not seen either during the previous study [14].

Adequate nutritional treatment in patients with COVID-19 appears to be an important element and would be recommended as a method of primary nutritional support and in critically ill patients [26,27]. Different guidelines state the importance of nutrition risk screening in critically ill patients with COVID-19. Most of the guidelines provide recommendations on the prescription of energy and protein in patients with COVID-19 using a predictive equation [28].

Older individuals and/or those with comorbidities, such as diabetes, obesity, cardiovas-cular disease, lung problems, and kidney and liver diseases, and with special nutrition needs seem to be more vulnerable to the pandemic. In parallel, quarantine, which has been applied in most countries as a measure to reduce the transmission of COVID-19, may affect dietary habits, with a trend toward more “comfort” energy- or carbohydrate dense foods (stress-eating) and increase body weight. The prolonged duration of such behaviors, if combined with reduction in physical activity, may negatively influence health status [29].

All patients with COVID-19 should undergo a nutritional study when being admitted into hospital, especially if they are over 60 years of age, which is the group with the highest risk of mortality, as well as those with comorbidities or immunosuppressive diseases. Protein intake is crucial to maintain and prevent the loss of fat-free mass (muscle) and for the synthesis of antibodies and immunoglobulins, which are so necessary to cope with coronavirus. Therefore, the use of high-protein formulas is recommended [30]. It can be seen from the study that the increase in the intake of high-protein diets increases compared to previous years in 2020 and 2021.

According to the ESPEN 2020 [28] guide, protein requirements are generally estimated using formulas such as:1.0 g/kg/day in older people; the amount should be adjusted individually with respect to nutritional status, level of physical activity, disease status and tolerance.1.0 g/kg/day in polymeric hospitalised patients to prevent body weight loss, improve functional outcome and reduce the risk of complications and hospital readmission. In critically ill patients, they recommend 1.3 g/kg/day.

The ASPEN Guidelines [29] also recommend the use of high-protein formulas to achieve a target of 1.2–2.0 g/kg/day. Not only the use of hyperproteic but also the use of hypercaloric enteral nutrition has increased compared to previous years. Based on the latest ESPEN clinical guidelines on nutritional support in the critically ill patient, in the case of complete enteral nutrition, the use of high calorie density (>1.25 kcal/mL) and high-protein (>20% total caloric value) formulas is recommended [28]. There is no consensus on the use of diabetes-specific or omega-3-enriched formulas. In ICU patients, one of the most used nutritional supplements has been diabetes-specific formulas (hypercaloric and hyperproteic), probably in relation to the hyperglycinaemia associated with high doses of corticosteroids used in these patients [31] and stress.

There has been a considerable increase in the prescription of dietary therapies in the different groups, but as mentioned in the introduction, the elderly are the patients in whom consumption has soared. The type of diet that has been prescribed most in this group is hyperproteic dietotherapeutic drugs, which therefore reflects that their use has been important in elderly patients throughout the evolution of the pandemic.

## 5. Conclusions

Therefore, it can be concluded that COVID-19 apparently had a significant impact on the prescription of nutritional support throughout the pandemic period, with a change and a positive trend in the age groups under 75 years, which in 2020 were less marked, and nutritional support could be one of the pillars of treatment with in-hospital and home use.

## 6. Limitations

The limitation of the study is that prescription of enteral nutrition to patients admitted in hospital and the prescription they receive once they have been discharged are outside the scope of the present study; these could have helped to explain more accurately the nutritional trends at home.

## Data Availability

Data was obtained from prescriptions of the hospitals of the National Health System. The data source used was the System for Analysis of the pharmaceutical provision of the Community of Madrid (Farm@drid).

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
