# Peer review of "Impact of COVID-19 on Short- and Medium-Term Prescription of Enteral Nutrition in the General Population vs. Older People in the Community of Madrid, Spain"

_nutrients, 2022, doi:10.3390/nu14193892_

Round 1

Reviewer 1 Report

Impact of COVID-19 on short- and medium-term prescription 2 of enteral nutrition for older people in the Community of Madrid, Spain.

Thank you for submitting your work and providing me with the opportunity to review your work, I hope you find my comments both helpful and constructive as that is my aim.

Unfortunately, there are minor grammatical errors throughout your paper, some have been highlighted below, but not all.

Your title suggests a focus on older people, but actually you completed a population based study, this should be reflected in your title.

Abstract

- in the first sentence the words 'the years' are not required

- it is unclear if the 'community of' is required throughout the paper, as at first I thought your work was applied to only community settings, but this does not appear to be the case, so perhaps this phrase needs to be removed

- clearer descriptions of peaks and which were most pronounced is required as this is currently difficult to follow

Introduction

- further information the relationship between enteral nutrition and COVID-19 is required, rather than there is a lack of data on the prescription of enteral nutrition during the pandemic

Materials and methods

- You state 'The data reflected belong to the...', which is unclear, do you mean the data was derived from...

- information on the type of supplement needs to be included, and what is meant by packs to support the readers understanding of your results

- the study does not respect the confidentiality of the data, the data was anonymised to ensure confidentiality of the participants

Results

- it is unclear what you are referring to when you state 'packs'

- Figure 1, why is there 2019 data included in this figure, this is not discussed within your text

- there is no information prior to your results on exploring diabetic products, this needs to be introduced both in your introduction and your methods

Discussion

- surely the previously published information on the peak of enteral nutrition due to COVID-19 in Madrid should have been introduced in the introduction

- further structure of your discussion is required, you provide an overview of the setting, and then in the following paragraph the impact of this on enteral nutrition, when this is one paragraph

- there are a number of paragraphs with one or two sentences, these are not full paragraphs

- there are a number of errors in the academic writing of the discussion such as 'because of the latter' and 'as already seen'

Limitations

- there needs to be a separate section on the limitations of your study

Author Response

Dear reviewer,

Thank you very much for your comments, they were really useful. We have made some changes and I hope to address them all in this notes.

- Your title suggests a focus on older people, but actually you completed a population based study, this should be reflected in your title: the title has been adapted so that it is more accurate regarding the aim of the study.

- in the first sentence the words 'the years' are not required: removed.

- it is unclear if the 'community of' is required throughout the paper, as at first I thought your work was applied to only community settings, but this does not appear to be the case, so perhaps this phrase needs to be removed: Community of Madrid is the name of the region of Madrid, if necessary because it would cause confusion, it can be removed, but as I said, it is the name of the territory. 

- clearer descriptions of peaks and which were most pronounced is required as this is currently difficult to follow: an image has been added to show what is understood for COVID-19 peaks and to better reflect the relationship between these and the increase in prescription.

- further information the relationship between enteral nutrition and COVID-19 is required, rather than there is a lack of data on the prescription of enteral nutrition during the pandemic: information has been added with more bibliography

- You state 'The data reflected belong to the...', which is unclear, do you mean the data was derived from...: yes, amended

- information on the type of supplement needs to be included, and what is meant by packs to support the readers understanding of your results: information added 

- the study does not respect the confidentiality of the data, the data was anonymised to ensure confidentiality of the participants: amended

- it is unclear what you are referring to when you state 'packs': changed to containers and cleared out what it is meant

- Figure 1, why is there 2019 data included in this figure, this is not discussed within your text: the reason has been added

- there is no information prior to your results on exploring diabetic products, this needs to be introduced both in your introduction and your methods: information has been added to introduce this

- surely the previously published information on the peak of enteral nutrition due to COVID-19 in Madrid should have been introduced in the introduction: information from previous published study has been also added and referenced here

- further structure of your discussion is required, you provide an overview of the setting, and then in the following paragraph the impact of this on enteral nutrition, when this is one paragraph: few details have been added

- there are a number of paragraphs with one or two sentences, these are not full paragraphs: paragraphs have been reorganized

- there are a number of errors in the academic writing of the discussion such as 'because of the latter' and 'as already seen': amended

- there needs to be a separate section on the limitations of your study: added

Thank you once again. 

Kind regards,

Carolina.

Reviewer 2 Report

I appreciate that the study is correctly described as a qualitative look at enteral nutrition during 2020-2021, compared to 2016. The importance of the report was not overrated. It promised only to provide a picture of the situation during the pandemic compared to 2016. 

My only comment is that this is an analysis of change in the number of prescriptions written for enteral nutrition, not in the change in actual consumption.  This must be clarified. Consumption and prescription are used interchangeably throughout the manuscript.  When referring to this study design, the correct term is "prescription".

Author Response

Dear reviewer,

I highly appreciate your comments and yputr review. It has already been ammended and the appropriate word "prescription" has been aligned during all the article. 

Thank you once again. 

Kind regards,

Carolina.

Round 2

Reviewer 1 Report

Impact of COVID-19 on short- and medium-term prescription of enteral nutrition for older people in the Community of Madrid, Spain.

Thank you for re-submitting your paper and engaging fully in the review process. 

I would suggest the removal of 'community of' from the title, as this is not how other countries would refer to the population of cities.

There remains grammatical and sentence errors in your paper, especially in the amendments that need to be addressed.